# Comparison of Transcriptional Signatures of Three Staphylococcal Superantigenic Toxins in Human Melanocytes

**DOI:** 10.3390/biomedicines10061402

**Published:** 2022-06-14

**Authors:** Nabarun Chakraborty, Seshamalini Srinivasan, Ruoting Yang, Stacy-Ann Miller, Aarti Gautam, Leanne J. Detwiler, Bonnie C. Carney, Abdulnaser Alkhalil, Lauren T. Moffatt, Marti Jett, Jeffrey W. Shupp, Rasha Hammamieh

**Affiliations:** 1Medical Readiness Systems Biology, Walter Reed Army Institute of Research, Silver Spring, MD 20910, USA; saissesh@gmail.com (S.S.); ruoting.yang.civ@mail.mil (R.Y.); stacyann.m.miller.civ@mail.mil (S.-A.M.); aarti.gautam.civ@mail.mil (A.G.); leanne_detwiler@yahoo.com (L.J.D.); marti.jett-tilton.civ@mail.mil (M.J.); rasha.hammamieh1.civ@mail.mil (R.H.); 2The Geneva Foundation, Walter Reed Army Institute of Research, Silver Spring, MD 20910, USA; 3Firefighters’ Burn and Surgical Research Laboratory, MedStar Health Research Institute, Washington, DC 20010, USA; bonnie.c.carney@medstar.net (B.C.C.); abdulnaser.alkhalil@medstar.net (A.A.); lauren.t.moffatt@medstar.net (L.T.M.); jeffrey.w.shupp@medstar.net (J.W.S.); 4Department of Surgery, Georgetown University School of Medicine, Washington, DC 20057, USA; 5Department of Biochemistry and Molecular & Cellular Biology, Georgetown University School of Medicine, Washington, DC 20057, USA; 6The Burn Center, MedStar Washington Hospital Center, Washington, DC 20010, USA

**Keywords:** superantigens, gene expression, transcriptional dynamics, staphylococcal enterotoxins, SEB, SEA, TSST-1, toxins, biological networks, clustering, functional pathways, time–course analysis, cDNA microarray, human melanocytes

## Abstract

*Staphylococcus aureus*, a gram-positive bacterium, causes toxic shock through the production of superantigenic toxins (sAgs) known as Staphylococcal enterotoxins (SE), serotypes A-J (SEA, SEB, etc.), and toxic shock syndrome toxin-1 (TSST-1). The chronology of host transcriptomic events that characterizes the response to the pathogenesis of superantigenic toxicity remains uncertain. The focus of this study was to elucidate time-resolved host responses to three toxins of the superantigenic family, namely SEA, SEB, and TSST-1. Due to the evolving critical role of melanocytes in the host’s immune response against environmental harmful elements, we investigated herein the transcriptomic responses of melanocytes after treatment with 200 ng/mL of SEA, SEB, or TSST-1 for 0.5, 2, 6, 12, 24, or 48 h. Functional analysis indicated that each of these three toxins induced a specific transcriptional pattern. In particular, the time-resolved transcriptional modulations due to SEB exposure were very distinct from those induced by SEA and TSST-1. The three superantigens share some similarities in the mechanisms underlying apoptosis, innate immunity, and other biological processes. Superantigen-specific signatures were determined for the functional dynamics related to necrosis, cytokine production, and acute-phase response. These differentially regulated networks can be targeted for therapeutic intervention and marked as the distinguishing factors for the three sAgs.

## 1. Introduction

*Staphylococcus aureus* (*S. aureus*) is widely circulated in nature and carried by 25–33% of normal individuals in the anterior nares and skin [1,2]. The extreme penetrance of this bacteria and its ability to colonize skin, open wounds, and other surfaces makes it a serious threat in facilities that provide health care [3,4]. The myriad of exotoxins synthesized and secreted by *S. aureus* include the Streptococcal enterotoxins (SEs), such as SEA-SEE, SEG-SEI, SEK-SET, and SEY, and the toxic shock syndrome toxin (TSST-1). As SEA is the most common toxin in food poisoning, SEB is recognized for its potent toxicity as a biological weapon, and TSST-1 is known for being the causative agent of lethal toxic shock [5,6,7], they remain the primary focus of *S. aureus* toxins research [8].

Staphylococcal enterotoxins and TSST-1 are superantigens (sAgs) that bind as an intact molecule to the major histocompatibility complex II (MHC) and interact directly with the variable region of the beta chain of T-cell receptors (TCRs) without the need for processing or presentation by the antigen presenting cells (APC). These interactions activate T-cells, resulting in massive production of cytokines and chemokines, activation-induced apoptosis, and T-cell anergy [9].

The interaction of sAgs with immune cells and the ensuing pathogenesis have been well documented [10,11]. Previous work from our lab identified a set of genes in human peripheral blood mononuclear cells (PBMCs) that were expressed as early as 2 h post-SEB treatment [11] and played important roles in tissue repair, inflammation, and increased vascular permeability. Supporting studies reported SEB-induced proinflammatory mediators contribute to vasodilation, vascular leak, and edema [12,13,14].

The immunologic barrier raised by the skin is a concerted effort from different cell types. Keratinocytes, melanocytes, and Langerhans cells actively contributed to the innate immune response to sAgs [15]. We presently focused on the melanocytes, which are dendritic cells of neuroectodermal origin and an integral part of the epidermis [16,17,18]. The dendritic nature and strategic location of melanocytes in the epidermal layer of the skin allow for an ideal milieu to interact with the extra-skin environment and build response coordination among neighboring shallow skin cells.

The immunological responses of melanocytes have been attributed to their ability to express MHC molecules and other various adhesion molecules, including intercellular adhesion molecule (ICAM)-1 and vascular cell adhesion molecule (VCAM)-1 [19,20,21]. In addition, melanocytes can produce several cytokines, tumor necrosis factor alpha (TNF-α), and transforming growth factor beta (TGF-β1) with potential functions in phagocytosis and antigen processing and presentation [20,22,23]. The immunomodulatory cytotoxic properties of melanocytes were highlighted in a recent in vitro study, where melanocytes were exposed to *C. albicans* infection [24]. Despite the wide coverage of melanocyte research and the increasing knowledge of their role in immune response, minimal information is available about the role of melanocytes in response to sAgs. The selection of melanocytes for the present study was further justified by the differential host-sAgs responses that were essentially determined by the significant structural differences of the three sAgs, thus affecting their interactions with the host cells [8,25]. Major findings include the toxin-specific melanocyte response dynamics enabling the distinction of toxin pathogenesis; in particular, we elucidated later-stage molecular events that could have the potential for common or customized therapeutic targets for the three toxins of choice.

## 2. Materials and Methods

### 2.1. Cell Culture and Toxin Treatment

Normal human epidermal melanocytes (NHEM) and the reagents required for culturing the cells were purchased from Clonetics^®^ (Lonza, Walkersville, MD, USA). Cells were maintained in Melanocyte Growth Medium (MGM) BulletKit^®^ according to the supplier’s instructions (Lonza, Walkersville, MD, USA). Cell cultures were established at the recommended starting cell density of 10,000 cells per cm^2^ and maintained in 150 cm^2^ flasks at 37 °C and 5% CO_2_ in a humidified incubator.

SEA, SEB, and TSST-1 were purchased from Toxin Technology (Toxin Technology, Sarasota, FL, USA). The toxins were diluted from the stock solution to 25 µg/mL in the MGM growth media (Lonza, Walkersville, MD, USA). On the day of the assay, the cells were treated with the appropriate amount of toxins to reach a final concentration of 200 ng/mL. The toxins were inactivated by adding TRIzol (Invitrogen, Carlsbad, CA, USA) at 0.5, 2, 6, 12, 24, and 48 h post-exposure (p.e.). As controls, untreated melanocytes were grown in parallel and harvested at the same time points. Each time point for each toxin was represented by a single culture.

### 2.2. RNA Isolation

Total RNA was isolated with TRIzol reagent (Invitrogen, Carlsbad, CA, USA) using the manufacturer’s procedure, followed by a cleanup procedure using the RNeasy MinElute Cleanup kit (QIAGEN, Germantown, MD, USA). The integrity of the extracted RNA was assessed using the 2100 Bioanalyzer instrument (Agilent Technologies, Santa Clara, CA, USA), and RNA integrity number (RIN) values were recorded.

### 2.3. Transcriptomic Assay and Analysis

The dual dye microarray hybridization was carried out using the SurePrint 4 × 44 K v2 Microarray Kit (Agilent Technologies, Inc., Santa Clara, CA, USA) following the vendor’s protocol. Cy-5-labelled 200 ng of purified RNA was co-hybridized with Cy-3-labelled reference RNA (Agilent Technologies, Inc., Santa Clara, CA, USA) and bound to Agilent 4 × 44 k slides (Design ID: 026652). These arrays contain 41,000 unique probes targeting 27,958 Entrez gene RNAs. Following standard protocol, overnight hybridization at 55 °C was followed by a series of washes. The slides were scanned with an Agilent DNA microarray scanner and the features were extracted using the default setting of the Feature Extraction software (Feature Extraction software v.10.7, Agilent, CA, USA). The genes that displayed transcriptomic expressions at a fold change higher than 2 (fold change ≥ 2) were selected for further analysis.

Gene expression analysis used functions available in the Bioconductor Project [26] and functional heatmap tool (https://bioinfo-abcc.ncifcrf.gov/Heatmap/ (accessed on 26 August 2021). GeneSpring v.10.1 (Agilent Technologies, Inc., Santa Clara, CA, USA) was used for data visualization. Enrichment analysis was performed using Ingenuity Pathways Analysis (IPA, QIAGEN, Inc., Germantown, MD, USA). The data from this study was submitted to GEO under accession number GSE124756.

### 2.4. Gene Expression Validation by Nanostring Assays

A custom NanoString panel (NanoString Technologies, Seattle, WA, USA) was designed for genes deemed functionally important for the current study. The results and discussion section justify our choice of genes listed in Appendix A. Six genes—*GIGYF2*, *INO80*, *USF2*, *WDR89*, *PPIA*, and *EIF2B1*—were selected as housekeeping genes based on their stable expression levels in melanocytes [27]. We followed the standard nCounter instructions [28], a master-mix containing hybridization buffer, Reporter ProbeSet, and Capture ProbeSet (volume:volume ratio of 1:1:0.5) was prepared, of which 25 μL was added to 5 μL target RNA. The GEN2 Prep Station incubation time was set at the higher sensitivity setting (3 h) and 280 fields of view (FOV) were routinely captured. Analysis and normalization of the raw NanoString data was conducted using nSolver Analysis Software v3.0 (NanoString Technologies, Seattle, WA, USA).

## 3. Results

### 3.1. Genomic Responses to the Three Toxins Are Characterized by Unique Host Expression Patterns

Principal component analysis (PCA) of transcriptomic expression data showed time-resolved clustering patterns of melanocytes exposed to three toxins for six treatment sequels (Figure 1). PC1 and PC2 represented 21.7% and 16.1% of the total variance; thus, together, PC1 and PC2 represent nearly 38% of the total variance. Within the transcriptomic variance defined by PC1 and PC2, we found three distinct clusters for each of the toxin types. These time points emerged clustered following longitudinal trends. For example, 30 min and 2 h SEA p.e. time points clustered together, and this combination was labelled as the early treatment phase. The early treatment phase was distantly located in the PCA plot from the middle treatment phase and was defined by 6 h and 12 h SEA p.e. time points. Finally, the late treatment phase was defined by 24 h and 48 h SEA p.e. time points, which were juxtaposed in the PCA landscape and distally located from the middle phase. A hypothetical line connecting these three treatment phases showed a potential temporal trend. A very similar picture emerged from TSST-1. The genes responding to SEB treatment, however, showed a different clustering pattern, which was more apparent in the late treatment phase of SEB. A considerable Euclidian distance was observed between 24 h and 48 h SEB p.e. Therefore, unlike SEA and TSST-1, we included 24 h SEB p.e. in the middle treatment phase along with its original two members, namely 6 h and 12 h SEB p.e. This arrangement automatically labelled 48 h SEB p.e. as the sole candidate of the SEB late-treatment phase. Interestingly, the middle-to-late treatment phases (12 h, 24 h, and 48 h) of SEA p.e. clustered closely to the middle treatment phases (6 h, 12 h, 24 h) of SEB p.e.

Since neither of the time point experiments have technical or biological replicates, the present strategy of grouping time sequela into the early, middle, and late treatment phases essentially enhanced the statistical confidence of the overall results. Using the longitudinal patterns of transcriptomic expressions, we sub-grouped the genes in three sets: (i) the ‘Early’ gene group, in which the transcriptomic fold changes were greater than |2| for at least one of the two time points (30 m and 2 h p.e.) of the early treatment phase; (ii) the ‘Consistent’ gene group, in which the transcriptomic fold changes were greater than |2| in all time points, and (iii) the ‘Late’ gene group, in which the transcriptomic fold changes were greater than |2| for at least one of the two time points (24 h and 48 h p.e.) of the late treatment phase. The exception was the SEB treatment, for which the late treatment phase included only 48 h p.e. Next, we combined (i) and (ii) to form ‘Early–Consistent’ gene groups; similarly, (ii) and (iii) were combined to form ‘Late–Consistent’ gene groups. These gene groups were used for functional analysis.

Appendix A depict Early–Consistent gene profiles of SEA, SEB, and TSST-1, respectively. Likewise, Appendix A depict Late–Consistent gene profiles of SEA, SEB, and TSST-1, respectively. A total of 445, 123, and 376 transcripts emerged, and these time clusters were called ‘SEA—Early–Consistent’ (SEA-E), ‘SEB—Early–Consistent’ (SEB–E), and ‘TSST—Early–Consistent’ (TSST-1-E), respectively. As explained above, the clustering for late-phase SEB exposure was performed differently than late-phase SEA and TSST-1 exposures. Hence, genes responding exclusively at 48 h p.e. (for SEB, Appendix A) or in one of the two late p.e. phases (24 h or 48 h p.e. for SEA, Appendix A, and TSST-1, Appendix A) were combined with their respective consistently expressed genes (i.e., Appendix A for SEB, Appendix A for SEA, and Appendix A for TSST-1). A total of 555, 1071, and 661 genes emerged, and they were called ‘SEA—Late–Consistent’ (SEA-L), ‘SEB—Late–Consistent’ (SEB-L) and ‘TSST—Late–Consistent’ (TSST-1-L), respectively.

### 3.2. Differences in Transcriptional Regulation in Response to the Three Toxins

In agreement with the PCA trend, the number of genes showing altered transcription varied greatly in response to the three toxins (Appendix A). Comparisons of early and late genomic responses to each of the toxins showed differences that were at their maximum after SEB treatment in both up- and downregulated genes. The largest number of genes responding with fold change (FC > |2|, nearly 1100 genes) were observed in SEB-L, whereas nearly 100 genes showed FC > |2| in SEB-E. In contrast, SEA-E and SEA-L comprised the least number of genes with FC > |2|. Nearly 450 genes showed transcriptomic modulations at early time points and nearly 600 genes were modulated during the late time points. Treatment with TSST-1 toxin elicited a response somewhat like SEA p.e. Interestingly, there was a common trend among all three toxins: the number of perturbed genes increased with the progression of treatment time, indicating the transcriptomic storm typically augmented by this family of sAgs [29,30,31,32] (Appendix A).

### 3.3. Biological Networks and Functions That Were Differentially Regulated by the Three Toxins

Functional analysis was performed using the genes listed under SEA-E, SEA-L, SEB-E, SEB-L, TSST-1-E, and TSST-1-L, respectively, to elucidate the time-dependent, toxin-specific enrichment profiles of biological and canonical functions. Appendix A lists the top biological functions (*p* < 0.001) and canonical networks (*p* < 0.01) associated with the three early treatment categories, SEA-E, SEB-E, and TSST-1-E. The list was filtered to include only those biological processes which were significantly enriched and functionally relevant to cell survival and the defense and maintenance of skin cells. In a similar fashion, genes belonging to the late treatment phase were probed to generate a list of significant biological and canonical processes that were enriched due to the prolonged toxin exposure (Appendix A).

Table 1 lists the top biological pathways (*p* < 0.001) and canonical functions (*p* < 0.01) that represent melanocytes’ dendritic cell-like (DC like) or macrophage-like property. ‘Antigen presentation pathways’, ‘dendritic cell maturation’, ‘IL17 signalling’, and ‘chemokine signalling’, among others, emerged as the top functions that are related to melanocytes’ immunogenicity.

ILK signalling emerged as a significant network that was conserved between the early and late treatment phases in response to all three sAg. Functional annotation of the 36 genes (Appendix A) associated with the ILK signalling pathway demonstrated association with two cellular processes, namely the cell death and tight junction signalling. Other networks that responded in common to at least two toxins and were conserved throughout the time–course of the study include acute phase response signaling, the antigen presentation pathway, the complement system, and agranulocyte adhesion and diapedesis.

The Venn diagram in Appendix A elucidated those biological networks that were common among as well as exclusive to SEA-E, SEB-E, and TSST-1-E. Nine networks related to cell survival and maintenance were affected by all three toxins. SEA-E and TSST-1-E shared the largest number (28) of networks, including those, which were associated with endometriosis, proliferation of connective tissue cells, and angiogenesis. SEA-E and SEB-E shared the smallest group of networks (2), which were related to skin disorders such as chronic skin disorder and chronic psoriasis.

A Venn diagram of the functional annotation enriched by the three late treatment phases, namely SEA-L, SEB-L, and TSST-1-L, (Appendix A), demonstrated a cohesive picture of the early treatment phase (Appendix A). The number of overall annotated networks was greater for the late phase (87 as compared to 66 networks for the early treatment phase), as described in Appendix A. The largest number of networks was shared between SEA-L and TSST-1-L as in the early treatment phase, with similar enriched networks, namely endometriosis and proliferation of connective tissue cells. A total of 19 networks were commonly enriched for SEA-L and SEB-L; hence, the late treatment phase was associated with a higher number of significantly enriched gene networks than those associated with the early treatment phase.

All three sAgs induced responses highly enriched for three biological processes: necrosis, skin diseases, and inflammation. Separate hierarchical clustering was performed using three gene sets, namely 217 genes from the necrosis network (Figure 2), 53 genes from the inflammation network (Appendix A), and 167 genes from the skin diseases network (Appendix A). The clustering analysis in Figure 2 identified four distinct groups of genes (indicated within yellow borders and labeled as groups A–D in Figure 2), which could be exclusive necrosis markers for TSST-1-L, TSST-1-E, SEA-L, and SEA-E, respectively. This hierarchical analysis failed to mine any exclusive signatures for SEB-E and SEB-L, respectively. Both the inflammation (Appendix A) and the skin diseases (Appendix A) clusters were mined as a single set, each under SEB-L (labelled group A in Appendix A, respectively). The complete list of all six gene sets is compiled in Appendix A.

### 3.4. Confirmation of Expression Pattern for Select Genes from the Necrosis Clusters

We performed validation of gene expression levels by NanoString nCounter^®^ technology. Appendix A lists the top thirteen highly perturbed genes (up- and downregulated) grouped under necrosis. This list is limited to genes responding only to SEA and TSST-1 for two reasons: first, none of the genes responding in the SEB-E phase were grouped in the three clusters discussed above (Appendix A), and second, a lack of sufficient RNA samples for the SEB 48 h treatment point forced us to exclude genes that belong to the SEB-L treatment phase.

Overall, a positive correlation was observed between the NanoString and microarrays results. Of the 13 genes tested, 12 genes followed the same directionality of fold changes for the NanoString and the microarray results (Appendix A) with the exception of one gene (PLCB1).

## 4. Discussion

The present study investigated in vitro host gene expression patterns induced by SEA, SEB, and TSST-1 during six time points ranging from 0.5 h to 48 h post-toxin exposure. A less frequently tested human skin cell type, but a major component of skin cell-mediated immunology, namely melanocytes, were selected as the target cells. The hybrid character of melanocytes was highlighted as we mined those biofunctions which were linked to the dendritic cell activities and/or the macrophage-based immune responses. This study could have benefited from incorporating additional time points to enhance the resolution of sequential biological events. For instance, our data suggested that the dosages of SEA and TSST-1 used for melanocyte treatments were potentially exhaustive within 24 h p.e.; in this context, extended time points could be highly informative. Furthermore, additional replicates in this study would result in better statistically significant gene identification. To mitigate this drawback to some extent, we mined the networks that met the cut off *p* < 0.05 using hypergeometric tests.

### 4.1. Distinct Temporal Trend of Pathogenesis Initiated by sAgs

The three toxins SEA, SEB, and TSST-1 of the sAg family are distinct in their structural, functional, and mechanistic properties [7,8,33]. Present literature not only lacks an understanding of molecular pathogenesis underlying the sAgs’ toxicity, but also fails to fully comprehend the role of melanocytes in response to sAgs. The melanocytes’ dendritic-like nature and their strategic location in the superficial layers of skin qualify them to be excellent mediators of initial immune defense against the sAgs [16,17,18,19,20,21,22]. We presented a whole genome-level investigation to compare the melanocytes temporal responses to SEA, SEB, and TSST-1.

A striking observation when comparing SEA and TSST-1 was the similarity in their gene expression patterns across the p.e. time course. Although SEA and TSST-1 share weak overall structural homology, TSST-1 can be displaced by SEA due to shared MHC class II binding sites [33]. This sheds light on the similarities in their mode of action as evidenced by the maximum number of shared networks for both early and late treatment phases.

Compared to SEA and TSST-1, the magnitude of transcriptional response perturbed by SEB was relatively smaller during the early treatment phase. However, the number of genes perturbed by SEB sequentially ramped up. This sort of delayed response is typical for any tissue that is not enriched with lymphocytes, as they are not the direct cellular targets of SEB [14]. Subsequently, SEB caused considerable genomic perturbations between 24 h and 48 h p.e. This trend is to be expected, as SEB typically causes a rapid neutrophil cell death accompanied by vascular congestion and leakage 24 h p.e., causing a shift to a predominantly adaptive immune response [30]. A perturbation in eNOS signalling pathways, potent vasodilators, was reported in the current study.

Another important observation was the temporal differences between SEA- and SEB-induced pathogenesis, particularly during their middle-to-late treatment phases (Figure 1). Nevertheless, a certain cohesiveness emerged between these two sAgs at the functional level. There were 11 and 19 networks that were synchronously enriched by both SEA and SEB at the early and late treatment phases (Appendix A). This fact may demonstrate an underlying similarity in their mode of pathogenesis. Early pathogenesis caused by SEA- and SEB-perturbed genes manifested in skin disorders. In concurrence, SEB exclusively targeted genes linked to T-lymphocytes and their related functions, whereas SEA targeted glucose and protein metabolism networks. The consequences may include dysregulation of immune functions, apoptosis and cell death.

All three toxins enriched several networks related to cell death at early exposure phase and this response continues throughout the time course of the study. This response could be attributed to the moderately high doses of toxins used in the present study. Even though the three toxins perturbed the similar networks during the early exposure phase, as time progressed, each toxin had its unique mode of action in achieving the outcome manifested by cell death and apoptosis. One of the networks that was consistently perturbed by all three toxins across the p.e. time-course was ILK signalling. ILK functions as a kinase and signal transmitter or as a scaffold protein to facilitate cell–matrix interactions, cell signalling, and cytoskeletal organization [34]. These signals control processes related to survival, proliferation, differentiation, adhesion, migration, contractility, and neovascularization. Inhibition of ILK arrests the cell cycle and promotes apoptosis [35]. This is a key observation to support the following argument.

Early perturbation of genes associated with superoxide radical degradation in SEA indicates an oxidative stress-driven early onset of cell death [36]. TSST also perturbed this mechanism at later time points. Treatment with SEA and TSST down regulated the transcriptional levels of SOD1 and TYRP1, which potentially diminished the synthesis of different isoforms of superoxide dismutase (SODs). The potential loss of SODs highlighted the onset of oxidative stress initiated by the toxins [37], ultimately leading to onset of apoptosis during the late phase p.e.

Additional aspects of the apoptotic network, such as ERK/MAPK, were enriched by SEA and SEB at early p.e. phases, which appears to show a SE-induced apoptotic pathway distinct from that induced by TSST-1 [38,39]. We observed increased expression levels of FOS and NFAT genes during early p.e. SEA and SEB treatments. The FOS gene encodes the proto-oncogene c-FOS protein and NFATs, which are known widely for their cytokine gene expression properties and have been increasingly shown to regulate other genes related to cell cycle progression, cell differentiation, and apoptosis [39,40]. Late phase, SEB p.e. up-regulated genes that encode oncoproteins, such as Rho GTPase, which is also linked with ERK/MAPK [41]. Consequently, the G2/M DNA Damage Checkpoint Regulation, a critical biofunction closely linked with apoptosis, was highly perturbed. At late p.e. phase, SEA cross activated PI3K/AKT signaling, a critical pathway which affects many intracellular processes, including cell survival, growth, and migration.

### 4.2. Late Phase SEB Is Associated with Certain Dermatological Disorders

sAgs have long been implicated in the development of various inflammatory skin diseases such as psoriasis, atopic dermatitis, Kawasaki Syndrome, etc. [42,43] We observed that all three toxins modulated genes associated with the pathogenesis of psoriasis and chronic psoriasis starting from the early treatment phase. Psoriasis is often associated with functions like cell death, inflammation, autoimmune syndrome, and the production of ROS and nitric oxide [44]. From early to late treatment phases, SEA and TSST-1 shifted the expression of the gene enriching networks that are linked to lichen planus and endometriosis. During the late treatment phase, SEB regulated two unique set of genes that are closely linked to psoriasis and dermatomyositis, respectively. These genes are listed under their respective disease names in Appendix A. Concurrent enrichment of oxidative stress networks could be related to the NRF2-mediated oxidative stress response and eNOS signaling pathways. Together these networks typically compromise the host’s antioxidant defense mechanisms, a hallmark indicator of psoriasis [45].

### 4.3. Several Genes of Immunological Networks Are Differentially Modulated by Toxins

The skin exhibits a highly specialized innate immune response to invading pathogens and external stimuli. The major immune players—keratinocytes, Langerhans cells, dendritic cells, resident T-cells, and innate lymphoid cells—act in a coordinated fashion, from sensing the external stimuli to communicating through inflammatory signalling cascades, to ultimately regulating immune homeostasis [46,47]. Accumulating evidence uncovered a hybrid role of melanocytes in regulating innate and adaptive immunity [16,17,18,19,20,21,22,24,48,49,50,51,52,53,54]. Similar to keratinocytes, melanocytes express several types of toll-like receptors (TLRs) and have the ability to produce several pro-inflammatory cytokines and chemokines [48,52,54]. Melanocytes also regulate the adaptive immunity through their functional similarities to lysosomes, such as capability to phagocytose and their antigen presentation and processing aptitudes [20,48,55]. In this context, we listed those networks (Table 1) which are associated with melanocytes’ hybrid role in responding to sAgs.

All of the toxin-induced adaptive immune responses could be attributed to the networks associated with leukocyte (granulocyte/agranulocyte) adhesion, a marker for second tier responses to inflammation induced by infection. Although all toxins contributed to adaptive immunity simulation, the patterns of cytokine production and acute-phase responses differed among the three toxins. For instance, during the early treatment phases of both SEA and SEB, the cytokine and chemokine signalling networks were comprised of CXCL1, CXCL12, and PLCB1, which control leukocyte trafficking; CCL2 and CCL7, which are involved in monocyte migration and macrophage recruitment; and CFL1, which regulates cell morphology and cytoskeletal organization. Early host responses to SEB and TSST-1 included an acute phase response signal that typically triggers non-specific inflammation, leukocytosis, complement activation, protease inhibition, clotting, etc. These responses persisted until 48 h p.e.

All three toxins perturbed IL-17 signalling, a pro-inflammatory signal that bridges innate and adaptive immune responses by playing critical roles in T-cell activation and in promoting the expansion and recruitment of innate immune cells, such as neutrophils [56]. The IL-17 signalling pathway was implicated in response to toxins via alterations of the transcription of several genes in this network, including CXCL1, CXCL5, CXCL8, CCL2, CCL20, and MAP2K6.

## 5. Conclusions

To our knowledge, this is the first mRNA-level study describing the temporal response of human melanocytes to three staphylococcal superantigenic toxins, namely SEA, SEB, and TSST-1. We observed distinct temporal patterns of transcriptomic regulation for the three individual toxins. The majority of the identified networks were related to necrosis and inflammation, in agreement with previous publications [38,39,40], although most of the past studies targeted different cells than melanocytes. Pathways related to innate immunity, such as the patterns of cytokine production and acute-phase response, showed toxin-specific regulation. The time-resolved response to SEB assault took a more differential pattern than SEA and TSST-1. In conclusion, these three toxins followed distinguishable pathways to achieve a common endpoint manifested by the cell death coordinated with apoptosis and necrosis. Hence, the temporal knowledge of their pathogenesis could be the key to customized intervention.

## Figures and Tables

**Figure 1 biomedicines-10-01402-f001:**
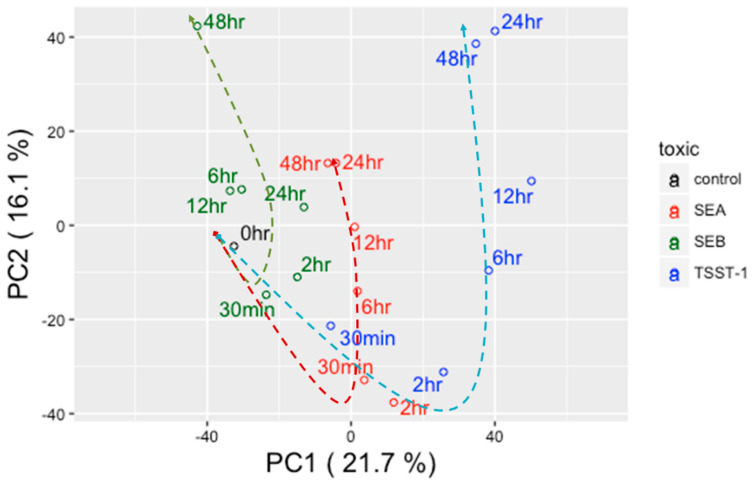
Principal components analysis (PCA) of time-resolved gene expression values. Black-, red-, green-, and blue-colored open circles represent control, SEA, SEB, and TSST-1, respectively. Dotted lines trace the temporal shifts caused by different toxins; here red, green, and blue dotted lines represent control, SEA, SEB, and TSST-1, respectively.

**Figure 2 biomedicines-10-01402-f002:**
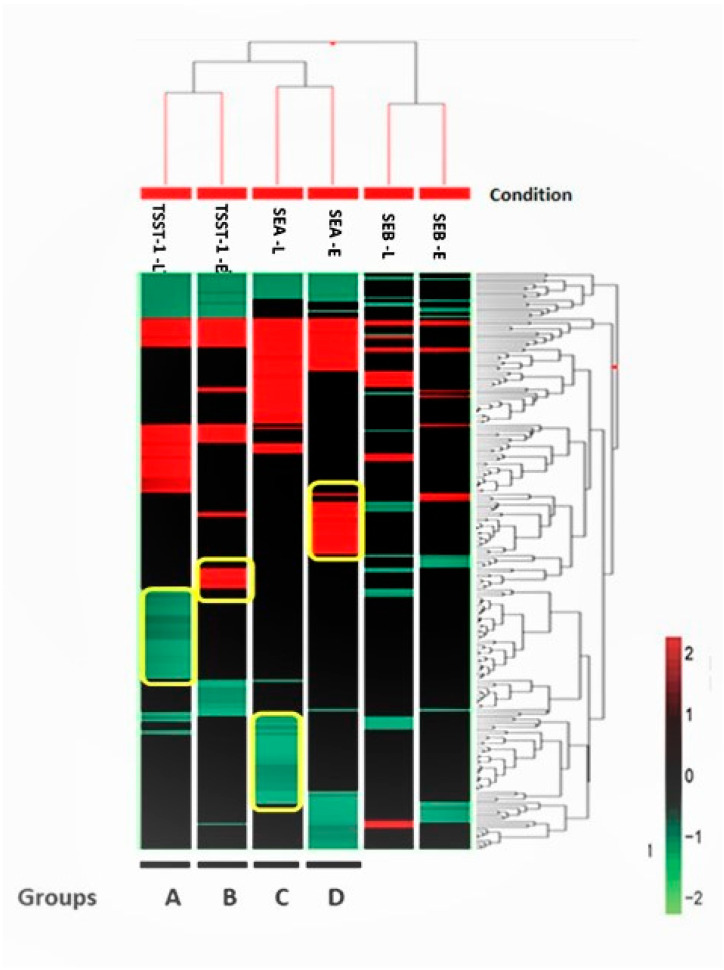
Hierarchical clustering analysis using of 217 genes with a log_2_ fold change > |2| enriching the necrosis pathway. The Euclidian algorithm was used to sort both conditions and genes. Each block represents one gene, and its color code is at the bottom right. Clusters bordered by yellow lines represent those genes which were potentially unique signatures of the particular condition. The conditions from left to right are named as TSST-1-L, TSST-1-E, SEA-L, SEA-E, SEB-L, and SEB-E, which represent TSST-1 at the late time point, TSST-1 at the early time point, SEA at the late time point, SEA at the early time point, SEB at the late time point, and SEB at the early time point, respectively.

**Table 1 biomedicines-10-01402-t001:** Biological pathways (*p* < 0.001) and canonical functions (*p* < 0.01) that represent melanocytes’ dendritic cell-like (DC-like) or macrophage-like property. Networks which are perturbed by the toxins are double tick (√√) marked. In addition, the association of the networks with DC-like and/or macrophage-like properties are noted by single tick (√) mark.

Biological or Canonical Functions	Toxin			Biofunction Relevant to Which Melanocyte Character?	
	SEA	TSST	SEB	DC-Like	Macrophage-Like
Early					
Adhesion of blood cells	√√	√√		√	√
Antigen Presentation Pathway	√√	√√		√	√
Cdc42 Signalling	√√	√√		√	
cell movement of leukocytes	√√	√√		√	√
cell movement of phagocytes	√√	√√		√	√
Chemokine Signaling		√√	√√		√
chemotaxis of phagocytes	√√			√	√
Complement System	√√			√	√
Crosstalk between Dendritic Cells and Natural Killer Cells	√√			√	√
Dendritic Cell Maturation	√√			√	√
Differential Regulation of Cytokine Production in Macrophages and T Helper Cells by IL-17A and IL-17F	√√			√	√
ERK5 Signalling	√√				√
HMGB1 Signalling	√√			√	√
IL-17 Signalling	√√			√	√
IL-17A Signalling in Fibroblasts	√√			√	
IL-8 Signalling	√√				√
Immune response of cells	√√			√	√
Immune response of leukocytes	√√			√	√
Immune response of phagocytes			√√	√	√
Inflammatory response			√√	√	√
MAPKKK cascade			√√	√	√
Migration of phagocytes	√√			√	√
Oxidative Phosphorylation	√√			√	√
PDGF Signalling	√√				√
Proliferation of immune cells	√√			√	√
synthesis of prostaglandin	√√			√	
synthesis of prostaglandin E2	√√			√	
T-cell lymphoproliferative disorder	√√			√	√
Late					
Activation of blood cells	√√	√√		√	√
Adhesion of blood cells	√√	√√		√	√
Aggregation of blood cells	√√			√	√
Antigen Presentation Pathway	√√			√	√
Autophagy of cells	√√			√	√
Cell movement of connective tissue cells	√√				√
Cell movement of leukocytes	√√			√	√
Chemokine Signalling	√√			√	√
Chemotaxis of neutrophils	√√			√	√
Chemotaxis of phagocytes	√√			√	√
Complement System	√√			√	√
Crosstalk between Dendritic Cells and Natural Killer Cells	√√			√	√
Dendritic Cell Maturation	√√			√	√
Differentiation of hematopoietic progenitor cells	√√			√	√
eNOS Signalling	√√			√	√
IL-17 Signalling	√√			√	√
Immune response of cells	√√			√	√
Immune response of leukocytes	√√			√	√
Metabolism of eicosanoid	√√			√	√
Metabolism of prostaglandin			√√	√	
Migration of antigen presenting cells			√√	√	√
Migration of phagocytes			√√	√	√
Phagosome Maturation			√√	√	√
PI3K/AKT Signalling		√√		√	
Signalling by Rho Family GTPases		√√		√	√
Superoxide Radicals Degradation		√√		√	√
Synthesis of prostaglandin		√√		√	
Transmigration of phagocytes		√√		√	√

## Data Availability

Data is contained within the article and supplementary files. Array data is available in GEO under accession number GSE124756.

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
