# Peer review of "Comparison of Transcriptional Signatures of Three Staphylococcal Superantigenic Toxins in Human Melanocytes"

_biomedicines, 2022, doi:10.3390/biomedicines10061402_

Round 1

Reviewer 1 Report

The manuscript reports important data on transcriptional signatures of three Staphylococcal super-antigenic toxins in human melanocytes. It requires minor revision before its acceptance for publication.

Abstract

Line 35, time-resolved

Introduction

Line 75, the manuscript is not “in-vitro”. In –vitro study described in …

Materials and Methods

Line 165, sub-grouped

Results

Line 239, observed for the late treatment phase?

Discussion

Lies 316 and 319: 48 h p.e.     signalling pathways, a potent

References

Please follow the journal’s formatting rules: Title NOT in italic, abbreviated journal name in italic, only journal volumes but no issues.

Line 492, p. 505-520

Several times, page numbers are missing, e.g. line 501, 503, 505, ..

Author Response

To,

The Editor and Reviewers

Special Issue of Biomedicines; “Omics Approaches to Immune-Mediated Inflammatory Diseases: Towards Novel Biomarkers and Potential Therapeutic Targets”

Subject: Responding to the Reviewers of the manuscript titled “Comparison of Transcriptional Signatures of Three Staphylococcal Superantigenic Toxins in Human Melanocytes

Dear Editor and Reviewers,

Thanks for reviewing our work and please see our responses to the reviewers’ comments in staring with NC.

All the comments were addressed and resolved. The page numbers were provided by the journal, so it is not my control. Finally, my DoD firewall stopped me downloading the MDPI EndNote file. Nevertheless, I changed the format to best match MDPI regulation. I can also share my EndNote file as needed.

Please let me know if you need anything else,

Sincerely,

Nabarun Chakraborty

Medical Readiness Systems Biology

Center for Military Psychiatry and Neuroscience

Room GW31, Building 503

Walter Reed Army Institute of Research, Silver Spring, MD 20910

Mobile Phone: 301-452-8940

Office Phone: 301-319-7363

Reviewer 2 Report

This is highly sophisticated study on transcriptome of SEA, SEB, TSST1, and suggested common and different actions to melanocyte. Although the study design does not correspond to actual biological situation, the findings may be of significance to further discuss the functional mechanisms of these toxins. Some comments are listed below.

  1. line 45: "Streptococcal enterotoxins (SEs) serotypes SEA, SEB, SEC1, SEC2,..." is not correct.  In this context, SEC subtypes should not be shown. "Serotype" is not appropriate now, because many SEs have been identified only genetically. This portion should be rephrased, for example, "staphylococcal enterotoxins (SEs) (SEA-SEE, SEG-SEI, SEK-SET, SEY), ....".  Additional information from this reviewer is that SElJ (staphylococcal enterotoxin-like protein J) is non-emetic protein, as well as TSST-1. Similarly, SElW, SElX, SElZ  (these are "enterotoxin-like protein") have been identified but their enterotoxigenic activity has not yet been demonstrated. ("L" representing "like" is attached.)
  2. Figure 1 and its explanations may be too difficult to understand for readers. General meaning of this analysis, i.e., the graph with PC1 and PC2 axes should be briefly shown somewhere. The detailed explanation may be omitted or shortened. 
  3. Table 2 is less meaningful. Consider to put to supplementary material. 
  4. Table 1 is located in Discussion, after Table 2, thus this order is strange. This Table 1 includes finding of Results, so consider to move to Results. Usually, Table/figure should not be added in Discussion, except for any case needing additional special explanation.
  5. Figure 3 is too simple and not understandable, less meaningful. Any more information may be added (full names of genes, or functional category of gene, etc.) Otherwise, it may be moved to supplementary materials.

Author Response

To,

The Editor and Reviewers

Special Issue of Biomedicines; “Omics Approaches to Immune-Mediated Inflammatory Diseases: Towards Novel Biomarkers and Potential Therapeutic Targets”

Subject: Responding to the Reviewers of the manuscript titled “Comparison of Transcriptional Signatures of Three Staphylococcal Superantigenic Toxins in Human Melanocytes

Dear Editor and Reviewers,

Thanks for reviewing our work and please see our responses to the reviewers’ comment.

I accepted and changed all the suggestion made here. In concurrence, I changed the other Figure/ Table legends as needed.

Please let me know if you need anything else,

Sincerely,

Nabarun Chakraborty

Medical Readiness Systems Biology

Center for Military Psychiatry and Neuroscience

Room GW31, Building 503

Walter Reed Army Institute of Research, Silver Spring, MD 20910

Mobile Phone: 301-452-8940

Office Phone: 301-319-7363
